# Single-Cell Radiation Response Scoring with the Deep Learning Algorithm CeCILE 2.0

**DOI:** 10.3390/cells12242782

**Published:** 2023-12-07

**Authors:** Sarah Rudigkeit, Judith Reindl

**Affiliations:** Section Biomedical Radiation Physics, Institute for Applied Physics and Measurement Technology, Universität der Bundeswehr München, 85577 Neubiberg, Germany

**Keywords:** single-cell response, phase-contrast, deep learning, cell survival, cell viability

## Abstract

External stressors, such as ionizing radiation, have massive effects on life, survival, and the ability of mammalian cells to divide. Different types of radiation have different effects. In order to understand these in detail and the underlying mechanisms, it is essential to study the radiation response of each cell. This allows abnormalities to be characterized and laws to be derived. Tracking individual cells over several generations of division generates large amounts of data that can no longer be meaningfully analyzed by hand. In this study, we present a deep-learning-based algorithm, CeCILE (Cell classification and in vitro lifecycle evaluation) 2.0, that can localize, classify, and track cells in live cell phase-contrast videos. This allows conclusions to be drawn about the viability of the cells, the cell cycle, cell survival, and the influence of X-ray radiation on these. Furthermore, radiation-specific abnormalities during division could be characterized. In summary, CeCILE 2.0 is a powerful tool to characterize and quantify the cellular response to external stressors such as radiation and to put individual responses into a larger context. To the authors knowledge, this is the first algorithm with a fully integrated workflow that is able to do comprehensive single-cell and cell composite analysis, allowing them to draw conclusions on cellular radiation response.

## 1. Introduction

Radiation has profound effects on eukaryotic cells, the fundamental units of life in complex organisms such as humans. When ionizing radiation interacts with biological tissues, it can cause significant damage at the cellular level, which can lead to tissue and organ failure, thus having an impact on a human’s health [1]. One of the primary types of damage is DNA damage, including single-strand breaks (SSBs) and more severe double-strand breaks (DSBs). DSBs, in particular, are highly cytotoxic and can lead to mutations, chromosomal rearrangements, or cell death [2]. Additionally, radiation can induce oxidative stress by generating reactive oxygen species (ROS) [3], disrupting cellular homeostasis [4], and damaging cell membranes [5], proteins, and other vital cellular components [6]. These effects can result in cell cycle arrest, apoptosis (programmed cell death), or senescence, altering normal cellular functions and potentially leading to diseases such as cancer [7].

To understand the mechanisms underlying radiation-induced cellular responses, it is important to perform basic radiobiological research [8]. The knowledge gained is essential for developing and improving radiation therapies for cancer treatment. A profound understanding of how different types of radiation affect various cancer cells can lead to more targeted and efficient treatments while minimizing damage to healthy tissues, such as in FLASH therapy [9] or particle minibeam therapy [10]. Furthermore, understanding the effects of radiation on a single-cell basis is vital for assessing the risks associated with radiation exposure [11]. Workers in nuclear facilities, astronauts, and patients undergoing radiation-based medical procedures benefit from this knowledge. By understanding the cellular responses to radiation, scientists can establish safe exposure limits and develop protective measures, ensuring the well-being of individuals exposed to radiation in various contexts [12].

The damaging effects of radiation and their impact on human health on a large scale have been known for more than a century [13]. Supported by technological developments, assays were developed that can be used to determine and quantify the reactions of large cell populations to radiation in vitro, such as colony-forming assays for cell survival [14], MTT assays for cell viability [15], and flow cytometry for assessing cell death [16]. These three assays are able to provide data that allows us draw conclusions on the major radiation effects that predominantly influence cellular integrity. Furthermore, cell survival, viability, and cell death can influence tissue functionality, immune reactions, and human well-being in cancer treatment and after radiation accidents. All the assays mentioned face the same challenge, as they can only be used at a particular time point for one endpoint in a sample, averaging over the reaction of thousands or millions of cells. This limits its applicability to low cell numbers, which are of interest, for example, in modern therapy methods such as micro- or minibeam therapy as well as in bystander research, where low cell numbers are irradiated. Furthermore, detailed information on how the cellular transition from the irradiated cell to a colony, a viable cell population, or cell death occurs is missing. Finally, post-irradiation treatments such as trypsinization for post-plating or color staining might interfere with the biological reactions. For example, cells that are already severely damaged might be lost during the trypsinization process, which is needed to perform the post-plating in the colony-forming assay. Post-plating is mostly necessary when accurate irradiation is needed for micro-beam [17,18] or bystander research [19]. In this case, the sample processing would alter the results. Finally, to be able to create a full picture of radiation damage, several assays have to be used, each on separate samples, and only indirect conclusions can be drawn, which limits the generalization of the conclusions drawn.

Therefore, since the mid-20th century, the single-cell response has been of key interest for researchers [20]. Only the limited amount of data storage and the long time needed for analysis made single-cell tracking unattractive for gaining knowledge. Increasing data storage, image analysis methods, and computational speed opens the door for performing complex analysis on single cells by tracking them through several generations.

To overcome the problem of slow result generation and therefore the limited use of single analysis, the CeCILE (Cell classification and in vitro lifecycle evaluation) project was started. In 2021, we published the first machine learning-based program that was able to detect and classify up to 100 cells in live-cell phase-contrast videos, CeCILE [21]. We also showed the promising perspectives that the use of artificial intelligence-based analysis provides for investigating cellular radiation responses. The aim of the current study was to generalize CeCILE to more cell lines and increase performance. The first step was to overcome the limitations, which come when there is a maximum number of cells that can be analyzed. The key developments in the new version, CeCILE 2.0, were to implement human-supervised cell tracking, which allows to track cells and their descendants over several generations and to generate cell lineages for each cell from the tracking data. This allows for the scoring of defects in the cell cycle, abnormalities occurring through division or throughout the cell cycle, as well as proliferation and cell survival. These developments represent a big step toward reaching the overall goal of the CeCILE project, which is to provide an open-source platform for automated analysis of cellular reactions imaged in the normal environment without interfering with the cells.

## 2. Results

### 2.1. Single-Cell Tracking

#### 2.1.1. Object Detection

In CeCILE 2.0, object detection was optimized for an unlimited number of cells and better accuracy compared to the first version [21]. Two videos from the test and tracking video datasets were used to test the performance of object detection. The video selection for one video with irradiated and one with non-irradiated cells was chosen to reflect the two extremes in terms of living and dead cells and final cell density. This helps to gain an overview of the general performance of the algorithm. The videos were fed into CeCILE 2.0 object detection. To quantify the performance of the algorithm, the evaluation of CeCILE was compared to the groundtruth data. The groundtruth was produced by manually labeling the data, as the human expert is commonly accepted to be the benchmark. In the performance test, both bounding box localization and classification were tested. The localization was tested by calculating the precision, recall, and F1 score as described in the Section 4. The results are shown for irradiated cells in Figure 1A and non-irradiated cells in Figure 1B. The object detector achieved a higher recall than precision in almost all frames of the irradiated cell video. The recall was between 0.82 and 1.0, with an increase to 1.0 more often in the first 200 frames, while the recall was around 0.95 after 200 frames. The precision was between 0.78 and 0.98 and followed the course of the recall. In the video of the irradiated cells, the object detector achieved an average precision over all frames of 0.88, an average recall of 0.93, and an average F1 score of 0.90. A better localization performance was achieved by the object detector in the non-irradiated video. Here, the recall was between 0.95 and 1.0 and the precision between 0.80 and 0.92. In the first frames, frame 0 to frame 20, between frame 140 and frame 160, and from frame 180 to frame 340, the precision decreased slightly, while the recall remained at a high level throughout the video. The average recall over all frames was 0.96, the average precision was 0.88, and the average F1 score was 0.92. The accuracy values for both videos are summarized in Table 1.

Therefore, the precision achieved was the same as for the video of irradiated cells, while a higher recall and F1 score were achieved for the video of unirradiated cells.

The mean average precision (mAP) score was used for classification evaluation. The mAP scores for each frame in the two videos are shown in Figure 1C for the video of irradiated cells and in Figure 1D for the video of unirradiated cells. In both videos, the object detector obtained the highest mAP values for the living cell class, the second highest for the dead cell class, followed by the round cell class. The lowest mAP values were obtained for the cell division class. In the irradiated video, the mAP liv started to be the lowest at about 0.90 and increased to about 0.95 after 50 frames. The mean mAP liv was 0.95; this is an increase of 6% compared to the first version of CeCILE (mAP liv = 89.15%) [21]. The mAP dead was highest at the beginning of the video, with values around 0.85. It was slightly lower between frame 300 and frame 456. The mean mAP dead rate was 0.82. In most frames, a mAP round of about 0.7 was achieved. The average mAP round was 0.72. Cell divisions did not occur in most frames. Therefore, the detector detected a few cell divisions. The average mAP div was 0.5. Overall, the mAP values are slightly better when compared to the first version of CeCILE [21], proofing the overall very good performance of CeCILE 2.0.

Overall, the algorithm shows very good performance. Nevertheless, looking at the data shows certain cases where the detection fails, especially when cells move over each other or are getting close together (see Appendix A). As in CeCILE 2.0 object detection, no time information is given; the algorithm cannot use the information from previous or later timepoints, which would help to increase performance. The use of 3D information in detection would help to better identify the position of a single-cell and differentiate between cells at different heights, but it would also increase complexity to a non-acceptable level at the moment.

#### 2.1.2. Cell Tracking

After object detection was performed for each frame, centroid tracking was applied as described in the Section 4. The accuracy of tracking was tested by comparing the tracking IDs predicted from the CeCILE 2.0 centroid tracker with tracking IDs from a manually labeled groundtruth on groundtruth bounding boxes in the two test videos. In the irradiated video, 97.77% (30,427 out of 31,120) of tracking cases were correct, and in the sham irradiated case, 98.51% (26,544 out of 26,946). This method can very well track the movement of cells but is not able to identify a cell division and assign two daughter cells to a mother cell. Therefore, to correct for incorrect classification, a manual supervising step is implemented in the CeCILE 2.0 workflow. The correction steps are implemented between object detection and tracking and after tracking. This allows the user to manually correct the bounding boxes, box labels, and box IDs, ensuring 100% accuracy in the classification and tracking through several cell cycles. As a final step, cell lineages for each cell and its descendants are created. A schematic sketch of the current workflow is shown in Figure 2.

### 2.2. Characterizing Radiation Response

To test the human-supervised workflow and to demonstrate the potential of CeCILE 2.0 in a real application, an irradiation experiment was performed and evaluated. In this experiment, CHO cells (25,000 cells in 3 mL) were seeded into two μ-dishes and incubated for 24 h. One μ-dish was irradiated with 3 Gy of X-rays, and the other was sham-irradiated. Both μ-dishes were placed in the live-cell imaging setup of the microscope, and video recording with the microscope was started immediately after irradiation. The recording was performed for 4 days. The obtained videos belong to the test and tracking video datasets of CeCILE 2.0, which were also used before to evaluate the performance of CeCILE 2.0 in object detection and tracking. As a biological response, endpoints such as cell vitality, cell cycle duration, cell proliferation-related abnormalities, and the number of cell divisions and vital daughter cells, as well as cell growth and survival, were scored.

#### 2.2.1. Cell Vitality

Cell vitality was scored as a simple measure to assess radiation effects without the need for single-cell tracking. Here, only the number of cells per class in a single frame is necessary for evaluation. Since the cells are self-replicating, the recorded area is quickly completely filled with cells. For the sham cells, it took 67 h to fully cover the growth area, and for the irradiated cells, full coverage was achieved 90 h after irradiation. Figure 3a shows the frames at 0 h, 24 h, 48 h, 72 h, and 96 h after irradiation for each video. As can be seen, the cells in both videos started with 27 and 23 vital cells and show a reduced growth rate for the irradiated cells compared to the sham cells.

A detailed evaluation was performed with CeCILE 2.0 for the first 28.3 h on the sham video and for the first 38 h on the irradiated video. These time periods were chosen because the cells in both videos were able to divide three times and were still well distinguishable from each other until the end of these time periods, which is necessary for proper tracking of the cells. In Figure 3b,c, the number of cells in each cell state is shown separately for the irradiated sample (c) and the unirradiated sample (b). The majority of cells in both videos are in the living cell state (liv), followed by the dead cell state, the round cell state, and finally the cell division state (div). The living cells therefore show a similar progression to the vital cells, which are all cells in the cell states liv, round, and div, with an increased growth rate of the sham cells compared to the irradiated cells. In the irradiated video, there are more dead cells with 11.88 ± 0.11 (mean ± SEM) dead cells per frame than in the sham video with a mean of 7.93 ± 0.12 dead cells per frame. This difference is statistically significant with a *p*-value < 0.05 (two-sample *t*-test). During the first 2 h, both samples had similar numbers of round cells, ranging from three to eight round cells. After two hours, the number of round cells in the irradiated video remained almost constant until 15 h after irradiation, while the sham sample had almost no round cells (mostly between 0zeroand three round cells per frame) during this time period. Between 15 and 20 h, almost no round cells were observed in the irradiated sample, similar to the sham sample. After 20 h, the number of round cells increased in both samples. Cell divisions occurred twice as often in the sham sample, with 117 cell divisions in 28.3 h, compared to the irradiated sample, with 55 cell divisions in 28.3 h.

Figure 3d shows a comparison of vital cells for the irradiated and non-irradiated samples. While the number of vital cells remains constant for the irradiated cells for 8.25 h, the sham cells begin to proliferate immediately. The growth of vital cells in the sham sample increases exponentially after 3.25 h, resulting in 158 cells after 28.3 h. In the irradiated sample, the cells also show exponential growth. However, radiation reduced arrest and resulted in a slightly lower growth rate with 81 cells at 28.3 h and 123 cells at 38 h. These results show that even a simple evaluation of the data by object detection can reveal significant differences in the treatment responses of the samples. A closer look at how individual cells respond to irradiation is provided by a tracking-based evaluation.

#### 2.2.2. Cell Cycle

The tracking results were used to determine information about the cell cycle. Here, for each cell, which appears itself or together with its offspring on more than 300 frames of a video, a full lineage was created as shown in Figure 4. This was the first time that it was possible to automatically produce such lineages and, with this, be able to follow the single-cell reactions throughout several divisions. The cell lineages serve as the basis for the following quantitative analyses.

The first quantitative analysis of the lineages was the cell cycle arrest after irradiation. Figure 5a shows the time the cells needed to start the first division after treatment. In the unirradiated cell colony, cells needed on average (7.6 ± 0.8) h (±SEM), with the first division scoring 5 min after starting the imaging and the last division being at 12.9 h. In contrast, in the irradiated sample, there was a significant (*p*-value < 0.0001, two-sample *t*-test) delay, as on average, the cells divided after (12.0 ± 0.8) h. The first division took place after 5.2 h and the last division took place at 22.5 h. Therefore, this analysis clearly showed a cell cycle arrest of 4.4 h.

Figure 5b shows the duration of the first and second full cell cycles for the sham and irradiated samples. The duration of a cell cycle was defined as the time between two complete cell divisions of a cell and was calculated for every mother cell, the cells that were initially in the video and based on which the cell lineages were created. Consequently, the duration of the first cell cycle was the time between the first and second cell divisions. To determine the first cell cycle duration for a mother cell, the mean of the first cell cycle durations of the corresponding two daughter cells is calculated. Analogously, the duration of the second cell cycle per mother cell was the mean duration between the second and third cell divisions of the four daughter cells. The first cell cycles of the sham sample lasted an average of 11.8 ± 0.3 h and were as long as the second cell cycle, which lasted an average of 11.4 ± 0.5 h. The shortest cell cycle lasted only 6.7 h, and the longest cell cycle lasted 14.7 h. In the irradiated sample, the first cell cycles lasted an average of 12.0 ± 0.5 h, and the second cell cycles took 10.6 ± 0.5 h, which are as long as in the sham sample. The longest cell cycle was 14.4 h, and the shortest was 8.9 h. This measurement shows that the cell cycle durations of the first and second cell cycles were equally long and also equally long for both samples. This suggests that after the first cell division, no difference in the cell cycle duration could be observed. This is also supported by the overall mean cell cycle duration shown in Figure 5c. There is no difference between sham-irradiated control and irradiated cells, with a mean cell cycle duration of (11.8 ± 0.3) h in the sham sample and (11.4 ± 0.4) h in the irradiated sample.

#### 2.2.3. Cellular Abnormalities

In the next steps, the cells were scored for abnormalities during or after cell division. Here, the cell lineages were used, in which the number of daughter cells and the fusion of the cell membrane of two daughter cells were visually analyzed. Fusion occurs when the cell is unable to completely separate the cell membranes between the daughter cells during cytokinesis. After some time, the two daughter cells fuse to form a binucleate cell. This type of abnormality is often related to aberrations linking two chromosomes, which cannot be pulled into a single daughter cell. Therefore, chromosome bridges are still present, and the cell will not be able to divide. The second type of abnormality is cell division into more than two daughter cells. In this experiment, only two or three daughter cells were observed.

Both abnormalities were quantified in the sham-irradiated cell population and in the irradiated cell population, as shown in Figure 5d. 35% of the cells in the irradiated sample show abnormalities, in contrast to only 4% of the sham cells. In the sham cell population, only one cell was divided into three daughter cells (cell 16, Figure 4a), and cellular fusion did not occur. In the irradiated sample, cellular fusion was 30% the dominant process. The abnormal number of daughter cells occurs in 9% of all cells. Some cells, such as cell 12 (Figure 4b, two times fusion, three daughter cells), show more than one abnormality.

#### 2.2.4. Cell Proliferation

Proliferation is the ability of a cell to reproduce offspring and thus undergo cell division. It can be described by the maximum number of cell cycles a cell has undergone in a given period of time and the number of daughter cells produced by a cell. CeCILE evaluates both the maximum number of cell divisions and the number of viable daughter cells per mother cell in a time period specified by the user. For this experiment, the first 28.3 h after irradiation was chosen as the time period. Only cells that were in one of the vital cell states when they first appeared were considered. Figure 5e shows the percentage of cells that underwent a certain maximum number of cell divisions in 28.3 h. In both groups, most of the cells underwent two cell divisions: 71% of the cells in the irradiated group and 63% in the sham group. A maximum of three cell divisions were observed. 33% of the sham group and 4% of the irradiated group underwent three cell divisions. More irradiated cells than sham cells were divided only once: 21% of the irradiated cells and 4% of the sham cells. 4% of the irradiated cells and none of the sham cells did not divide. On average, the irradiated cells underwent a maximum of 1.75 ± 0.13 cell divisions, and the sham cells underwent 2.29 ± 0.11 cell divisions. The maximum number of cell divisions undergone by each cell is statistically significantly different between the sham group and the irradiated group, with a *p*-value < 0.001 (two-sample *t*-test). These results can be compared to the cell cycle duration analysis. Both analyses are in correspondence and suggest most of the cells undergo two divisions, with a slight shift to more divisions for the non-irradiated sample. This suggests that the difference in proliferative capacity in the first 28.3 h originates from the cell cycle arrest due to the repair of the induced damage.

The number of daughter cells per cell is shown in Figure 5f. Cells that were not in a vital state at the beginning of their appearance were excluded from the analysis. In both groups, the mother cells most frequently produced four daughter cells, namely 46% (irradiated) and 60% (sham). In the irradiated group, a maximum of five daughter cells were produced by 4% of the mother cells. Also, 4% of the mother cells produced no viable daughter cells and one viable daughter cell. 25% of the mother cells produced two daughter cells, and 17% produced three daughter cells. In the sham sample, only 4% of mother cells produced three daughter cells, which was also the minimum number of daughter cells. 4% of the mother cells produced six daughter cells, 7% of the mother cells produced seven daughter cells, and 22% of the mother cells produced eight daughter cells, which was also the maximum number of daughter cells produced in the selected time period. On average, the irradiated cells produced 3.1 ± 0.25 daughter cells, and the sham cells produced 5.1 ± 0.4 daughter cells. The difference in the distributions of the number of viable daughter cells per mother cell is statistically highly significant with a *p*-value < 0.00005 (two-sample *t*-test).

The measured values can be compared to the model of exponential cell growth. The growth rates can be determined by the cell numbers at certain time points. Here, exponential growth with continuous time is taken using the formula
(1)$$x\;\left(t\right)=x_{0}e^{pt}$$
with *x*(*t*) being the number of cells at time *t*, *x*_0_ the initial cell number, and *p* the growth rate. The growth rate is determined by using the single logarithmic representation of the function with $\ln\left(x\left(t\right)\right)=pt+ln(x_{0})$ and a linear fit of the data as shown in Figure 6. The fit was performed for the exponential phase, which started at 3.25 h in the sham sample and 8.5 h in the irradiated sample. For the fit, *x*_0_ was the number of cells at the corresponding starting times. For the non-irradiated cells, the growth rate is, $p_{sham}=0.0632\pm 1.6\times 10^{-4}$ and for the irradiated sample $p_{irr}=0.0506\pm 4\times 10^{-4}$. Using the growth rate, the number of divisions and the number of daughter cells per mother cell can be calculated.

The number of daughter cells per mother cell are calculated as ${DC}_{calc}=\frac{x\left(t\right)-x_{0}}{x_{0}}$. The number of divisions is the number of DC divided by 2, as in a healthy cell cycle, each cell divides into two daughter cells. Table 2 shows a comparison of the calculated numbers of daughter cells per mother cell (DC_calc_) and the cell divisions (Div_calc_) with the experimentally derived (DC_meas_ and Div_meas_) values.

Overall, the values derived from exponential growth theory, together with the cell numbers, are in very good agreement with the values determined from the cell lineages. This shows that the trained machine learning model gives very consistent results from basic to complex analysis, which additionally confirm the theoretical models of cellular growth.

#### 2.2.5. Cell Survival

Finally, a comparison is made between a conventional cell survival assay and the calculated cell survival from the data acquired using CeCILE 2.0. Here, a colony-forming assay was used, where a colony was defined as a minimum of 50 cells that originated from one seeded mother cell within 5 days. As an experimental comparison, the cell survival from Rudigkeit et al. [21] was used. Here, a survival fraction after 3 Gy X-rays of (55 ± 6)% was measured. The data from CeCILE 2.0 allow for an estimate of cell survival. In the sham sample, 25 out of 27 cells were able to proliferate following exponential growth. All of these cells showed a normal cell cycle and are expected to be able to go on with proliferation until nutrient supply and space limit exponential cell growth. These cells can be expected to be colony-forming cells, although due to close contact in the sample at earlier time points, no colonies could be differentiated. All of these 25 cells (92.5%) were able to divide into 4 or more daughter cells in the first 28.3 h; therefore, for the estimation of cell survival, the criterion of having 4 or more daughter cells after 28.3 h will be used to define a cell as surviving. In the irradiated sample, 11 out of 24 cells (45.8%), were able to reach this criterion. Survival is now defined as
$$SF=\frac{N_{\mathrm{sample}\geq 4,\mathrm{DC}}}{N_{\mathrm{sample},\mathrm{total}}}\frac{N_{\mathrm{sham},\mathrm{total}}}{N_{\mathrm{sham}\geq 4,\mathrm{DC}}}\cdot 100$$
where *N*_*sample*≥4,*DC*_ is the number of mother cells in the evaluated sample that divided into more than four daughter cells (DC), and *N_sample_*_,*total*_ is the number of mother cells in the sample. *N*_*sham*≥4,*DC*_ and *N_sham_*_,*total*_ are the number of mother cells with more than four daughter cells and the total number of mother cells in the sham sample, respectively. The sham sample has an SF = 100% and the irradiated sample has an SF of 49.5%. This value is in very good accordance with the survival measurement with the colony-forming assay, showing that the measurement of the first cell cycles already gives very good evidence for the future of the cell and its ability to survive.

## 3. Discussion

In this study, we presented a novel method to study the cellular behavior of single eukaryotic cells. It is based on observing the cells for several days after irradiation by live-cell phase-contrast microscopy and analyzing the obtained data with the human-supervised algorithm CeCILE (Cell classification and in vitro lifecycle evaluation) 2.0, which is based on artificial intelligence. The introduced algorithm can detect and track cells on microscopic videos and classify them into four cell states depending on their morphology. It is also capable of evaluating various cell cycle-related endpoints such as proliferation, cell cycle duration, cell cycle abnormalities, and cell lineage. The first version of CeCILE published in 2021 [21] was, to our knowledge, the first to present an artificial intelligence-based algorithm for analyzing cell response to radiation on live-cell phase-contrast videos.

As a first step, we implemented an object detection system based on artificial intelligence that is able to detect all cells on all frames of a live-cell phase-contrast video. The object detection in CeCILE 2.0 represents an upgrade to the previously published one, as it overcomes the limit of 100 cells that could be detected in the first version. Now there is no limit to the number of detectable cells, opening the way for increased statistics and deeper analysis. The object detection algorithm chosen is a pre-trained, faster RCNN with ResNet-101 as the backbone CNN. The dataset includes images from a total of 20 experiments with different setups to increase generalization and widen the window of possible applications. It includes three cell lines, CHO-K1, LN229, and HeLa, which are commonly used in radiobiology. These cell lines have three important characteristics that are necessary for detection and tracking. They grow unchanged when seeded at low densities without clustering. They adhere to flat surfaces and grow in a 2D-like fashion, usually without overlapping. The cells of these cell lines are easily distinguishable and can be followed throughout a video as long as the cell density is not too high. In principle, CeCILE 2.0 can be applied to any cell line that meets these requirements, but it has to be tested if further training is necessary before application.

Object detection performance was evaluated using two videos of the test and a tracking dataset derived from an irradiation experiment. In the first video, CHO-K1 cells were irradiated with 3 Gy of X-rays, and in the second video, cells were sham irradiated. For these two videos, we created a detection and tracking groundtruth that was used to evaluate the performance of CeCILE 2.0 in both tasks, detection and tracking. First, object detection was evaluated in terms of its performance in localizing objects. This is the most important part of object detection in CeCILE 2.0, since tracking relies on the correct localization of cells. Here, the F1 score was calculated. Correctly detected cells had box overlaps with IoU values > 0.5 of predicted boxes and groundtruth boxes. The video of the irradiated cells achieved a mean F1-score of 0.90 with a mean precision of 0.88 and a mean recall of 0.93. The video of the sham cells had a mean F1 score of 0.92, a mean precision of 0.88, and a mean recall of 0.96. Thus, the object detector performed slightly better on the sham video. Overall, this means that over 90% of the cells were accurately localized in both videos. For a low-contrast biological sample, this is a very good value, and we conclude that the performance is good enough to proceed.

In addition to localization, an object detector also classifies the detected objects. For object detection, localization and classification are typically evaluated in a combined score, the mAP score. The mAP score is 1 for perfect detection and classification. The mAP can be calculated for each class individually and as an average mAP over all classes. The object detector achieved the best results in the detection and classification of the liv class. This class had a mean mAP of 0.95 in the irradiated video and 0.97 in the sham video. In the irradiated video, 75.1% of the cells were in class liv and in the sham video, 85.6% of the cells were in class liv. The second-best results were achieved in the dead class, with mean mAP values of 0.82 (irradiated) and 0.69 (sham). There were more dead cells in the irradiated video, with 17.0% of the cells, than in the sham video, with 9.8% of the cells. The third highest mAP score was achieved in the class round, with 0.72 in the irradiated video and 0.60 in the sham video. 7.6% of the cells were in the class round in the irradiated video, and 4.2% of the cells, in the sham video. By far the least common class was div. Only 0.3% and 0.4% of the cells in the irradiated and sham videos, respectively, were in this class. The mean mAP values were also the lowest, at 0.5 in both videos. This shows that the cell detection gave better results for classes containing more cells. However, it is important to note that the mAP score is much more affected by a missing bounding box or an incorrect classification when there are fewer cells in the class being examined. This can be seen in the classes liv, round, and dead, where higher mAP scores were obtained when there were more cells in a class. In particular, class liv, which contained the majority of cells, had very high mAP scores above 0.95, indicating that the vast majority of cells were correctly detected and classified. The mAP score of a class was only below 0.8 when less than 10% of the cells belonged to the class. As shown in the classification with a simple CNN, the classification struggles with cells that transit from one class to another. In the cell cycle, cells stay in the liv class most of the time. The class round as a precursor to cell division occurs only for about 30 min and the class div can be observed only for about 10 min. We imaged the cells every 5 min. Therefore, the probability that a cell is in a transition state is higher for the round and div classes than for the liv class. Dead cells change their morphology after death. They may undergo self-digestion, as in apoptosis, or they may be digested by other cells. In these processes, dead cells disappear after some time. Therefore, dead cells are difficult to detect and classify for an algorithm, but also for a human expert annotator, if the death occurred 1 h or more ago. Misclassifications in cell state transitions do not affect the result after tracking, since it is not important whether a cell enters a particular cell state one frame earlier or later. Furthermore, it is not important to track dead cells for the whole observation time since the only important information, namely that a cell is dead, has already been received. Therefore, it can be concluded that object detection provides all the information needed for tracking and performs well enough to be passed to a tracking algorithm. In order to improve the classification performance, the next development step of CeCILE will include the time information, i.e., the state of the cell, several frames before and after.

As a tracking algorithm, we implemented the centroid tracker proposed by A. Rose-brock [22] and adapted it to the special requirements of tracking cells in phase-contrast videos. Furthermore, we implemented the IoU as a second feature to take into account box overlap when matching bounding boxes, which further increases robustness. Since the centroid tracker is a location-based tracker, it is well suited for cells on phase-contrast videos. Cells on phase-contrast videos hardly move between frames when a frame-to-frame distance of 5 min is applied during recording. An appearance-based tracker is less suitable for cell tracking because cells change their morphology during the cell cycle. Also, trackers that expect objects to move toward a specific target cannot be applied because cells in culture move in a more random walk fashion due to the homogeneous distribution of nutrients in the culture. The centroid tracker is a hard-coded tracker that only considers the information of the previous and current frames. It is designed for accurate tracking of objects where each object has a track but does not provide the ability to track across cell divisions where a track splits into two tracks. However, it can be used to track between cell divisions and manually assign the tracks of two daughter cells to a mother cell afterwards. The performance of the implemented tracker was tested, such as object detection before, on the two videos of the test and tracking datasets. We tested the tracker on the bounding boxes of the groundtruth in order to test only the performance of the tracker and not the previously tested detector. Since the centroid tracker cannot track across cell divisions, all ID assignments between cell divisions predicted by the tracker and given by the groundtruth were compared to calculate the tracking accuracy. The tracking accuracy is the percentage of correct ID assignments out of all assignments. The centroid tracker achieved an accuracy of 97.77% in the irradiated video and 98.51% in the sham video. This shows that the proposed location-based tracker is well suited for cell tracking.

The detection and tracking implemented in CeCILE 2.0 both give very good results when used separately. In combination, however, the errors made by each are amplified. Errors in detection, such as missing bounding boxes, lead to errors in tracking, since tracking relies on the bounding boxes of object detection. For the evaluation of phase-contrast videos on a single-cell basis, 100% accuracy is required, and no errors can be accepted as they would falsify the result. In particular, switching cell identities or prematurely stopping the track will lead to incorrect results when evaluating the cell cycle and creating accurate cell lines. Therefore, we implemented two manual monitoring steps. The first step was implemented after object detection to allow the user to add any missing bounding boxes. With these corrections, the tracking algorithm can be applied, which leads to very accurate results. In the second correction step, the tracking IDs can be adjusted. The tracks of mother cells and their daughter cells can be combined. In both correction steps, the class labels of the bounding boxes can be corrected if necessary. With this supervised approach, the user can easily evaluate phase-contrast videos with 100% accuracy and obtain precise information about each cell in the video. Although very high accuracy could be achieved in the supervised mode, this is one of the main limitations of CeCILE 2.0. In the future, human supervision should be eliminated from the workflow. To be able to do this, a different approach to tracking is necessary, which is able to track over a cell division, recognize the daughter cells as such, and track them separately. For this, it is necessary to use the full video information, adding the time domain as a parameter, and not sticking to the single time planes. This development needs substantial improvement of the underlying model. One possible solution could be the use of conservation tracking, which was proposed for use in life sciences [23,24], or the use of the Hidden Markov Model [25].

In the experiment shown in this study, cells were imaged for 4 days and analyzed for cell viability, cell cycle and cell cycle abnormalities, proliferation, and survival. It is possible to extend the monitoring to more than 4 days, but the evaluation with CeCILE 2.0 is limited by the density of the cells. Proper detection and tracking are only possible as long as the cells are still distinguishable from each other. We show that the cells are in exponential growth during the imaging period, except for a cell cycle arrest of 4.4 h for the irradiated sample. After the first division, the cell cycle is constant, with (11.8 ± 0.3) h in the sham sample and (11.4 ± 0.4) h in the irradiated sample.

The cell lines generated by CeCILE 2.0 provide deep insight into the evolution of each cell. For example, it can be seen here that some cells show abnormalities in their cell cycles. In the videos analyzed, fusions of two daughter cells and divisions into three daughter cells were observed. Combinations of both abnormalities were also observed. Overall, 35% of the irradiated cells showed cell cycle abnormalities and only 4% of the sham cells, with division abnormalities being the dominant process. The object detection CeCILE 2.0 and the counted number of cells could also be used to correctly calculate the number of divisions and daughter cells and to prove the assumed model of exponential growth with cell cycle arrest for the irradiated cells. Based on the data, cell survival could be predicted, which was in good agreement with actual measurements of the colony formation assay in the same setup [22]. By comparing the results obtained, it can be concluded that the difference in cell growth between irradiated and sham cells is mainly a result of cell cycle arrests immediately after irradiation and therefore delayed cell divisions. Furthermore, due to unsuccessful cell divisions, on the one hand because more daughter cells died and, on the other hand, because of cell cycle abnormalities.

## 4. Materials and Methods

### 4.1. Data Set

For the dataset used for training CeCILE 2.0, 20 videos of cells of three different cell lines (HeLa, CHO, and LN229) were recorded via live-cell imaging with a standard inverted microscope in our lab. An overview of the used data can be found in Appendix A. During recording, cells were kept in a stage-top incubator that ensures a healthy environment for the cells and enables continuous monitoring for up to 5 days. From the recorded videos, frames were chosen to be labeled and included in the dataset. In the first 13 videos, a distinct time interval of 20 min or 100 min was chosen between the labeled frames, and in videos 14 to 20, only frames were chosen to be included in the dataset where at least one cell was in the state cell division. Cells were imaged after different treatments, under different conditions, and with different imaging modes, resulting in heterogeneous videos that represent a wide range of experiments. Cells were seeded into three different containers that have different optical properties. The containers were coated with either gelantine or CellTak or left uncoated. Cell samples were irradiated or left unirradiated. For testing the performance of object detection and tracking on videos, two videos were labeled. To create the videos of the test dataset, CHO cells were seeded on two μ-dishes without coating. After 24 h of incubation, one μ-dish was irradiated with 3 Gy of X-rays, and the other one was sham irradiated. Both μ-dishes were placed in the live-cell-imaging setup of the microscope, and recording of the videos with the microscope started immediately after irradiation. The recording was performed for 4 days. For the groundtruth, 457 frames of the irradiated sample were labeled, corresponding to a time range of 38 h, and 341 frames of the unirradiated sample were labeled, corresponding to a time range of 28.3 h. The different numbers of labeled frames were chosen because the irradiated sample has a decreased growth rate compared to the sham sample. In the evaluated time ranges, the cells in both samples were able to divide three times and could still be tracked accurately as the cell densities were not too high. To create a groundtruth of these videos, the videos were labeled by the CeCILE 2.0 object detector and tracker, and the labels were manually corrected after object detection and tracking using VIA image annotator software [26].

### 4.2. Object Detection

A faster RCNN implemented in the TensorFlow object detection API was chosen as an object detection model. This model is easy to use and adapt to custom datasets, and it is very accurate in detecting many small objects on a crowded image [27]. The Faster-RCNN also shows these characteristics when applied to microscopy images of cells [28]. To save computational time, transfer learning with a pretrained ResNet-101 model trained on the COCO dataset [29] from the TensorFlow 2 Model Detection Zoo was used. For identification of the cell-specific appearance, classification, and location, CeCILE 2.0 was trained and fine-tuned on the specific dataset described in the Section 4.1, which was split randomly (75%/25%) into a training dataset and a validation dataset. Training was performed on the training dataset, and for fine-tuning, the object detector’s predictions were evaluated during training on the validation dataset. During this process, only the top 10 layers of ResNet-101 were trained, and the rest were frozen. For inspection during the training, the qualification scores were used, which are described in the next section. This way, the training process could be inspected and the parameters fine-tuned accordingly. The data preparation and training pipeline for faster RCNN is implemented as described by Rosebrock [30] and on the official website of the TensorFlow 2 object detection API [31]. In the training process, the following parameters were fine-tuned: number of epochs, learning rate, aspect ratios and scales of the anchor boxes, data augmentation, non-maximum suppression, localization loss weight, classification loss weight, and objectness loss weight. These parameters have been chosen because during the development process, it turned out that they had an influence on the model’s performance. The major influences were seen in the learning rate and data augmentation. The final parameters are shown in Table 3.

Training was performed on a NVIDIA GeForce RTX 2080 super graphics card (NVIDIA corporation, Santa Clara, CA, USA) with 8 GB of VRAM (video random access memory).

### 4.3. Qualification Scores

For object detection qualification, first the number of true positives, false positives, and false negatives were counted. True positives are all boxes that overlap a groundtruth box with an intersection over union (IoU) > 0.5. The intersection over union is determined by measuring how much the predicted bounding box overlaps with the groundtruth bounding box. False positives are all boxes that do not overlap with a groundtruth box with an IoU > 0.5 and are, therefore, falsely predicted by the object detector. An example of a false positive is a prediction of two boxes for one object. Objects not predicted by CeCILE 2.0 object detection are called false negatives. After that, the following scores were calculated for each frame:

Recall, also known as true positive rate, is the percentage of cells correctly identified for a class out of the total cells for this class:$$Recall=\frac{True\;Positives}{True\;Positives+False\;Negatives}$$

The precision is the percentage of cells correctly predicted for a class out of all cells belonging to that class:$$Precision=\frac{True\;Positives}{True\;Positives+False\;Positives}$$

The F1-score is the harmonic mean of precision and recall:$$F1=\frac{2\cdot Precision\cdot Recall}{Precision+Recall}$$

For the mAP score, the class labels of the boxes were also taken into account when assigning the boxes as true positives, false positives, and false negatives. Now, only boxes are true positive if the box overlaps with an IoU > 0.5 with a groundtruth box and the label is the same. If the label does not match the label of the corresponding groundtruth box, the box is false positive. The boxes were sorted according to their confidence score that was assigned by the object detector, starting with the highest confidence score and going to boxes with smaller scores. The precision and the recall were calculated by taking only the current box and all boxes with a higher confidence score as one of the current boxes into account. This is repeated for all boxes, and in every step, one more box is taken into account. The average precision is the area under the resulting recall-precision curve. The mean average precision score was calculated for each class individually as the mean of the average precision for IoU < 0.5 (mAP liv, mAP round, mAP div, and mAP dead). If a class did not appear in the predictions and the groundtruth, it was assigned a mAP score of 0. The overall mAP was defined as the average of all classes present in each frame.

### 4.4. Centroid Tracking

The bounding boxes obtained from the detection are passed to a tracking algorithm. By implementing a tracking algorithm that uses the bounding boxes obtained from object detection, CeCILE 2.0 is able to track each cell throughout the video. The tracking algorithm assigns a unique ID to each bounding box in the first frame of a video. In the next frame, the tracking algorithm matches the bounding boxes with the bounding boxes of the previous frame. If a matching bounding box is found, it is given the same ID as its matching partner. If there is a bounding box in the current frame that has no matching partner in the previous frame, it receives its own unique ID. This matching process is repeated for all frames in the video.

To track the cells based on their location and the bounding boxes obtained from the detection, the centroid tracker developed by Adrian Rosebrock [22] was implemented in CeCILE 2.0 and adapted to the specific needs of cell tracking. This object tracker uses the OpenCV library in Python. In the first step of the centroid tracker, the centroids of each bounding box in a frame were determined by calculating the center coordinates of the bounding box based on the box coordinates:$$centroidX=\frac{startX+endX}{2}$$
$$centroidY=\frac{startY+endY}{2}$$
where *centroidX* and *centroidY* are the coordinates of the center of a bounding box in the x and y directions, *startX* and *startY* are the x- and y-coordinates of the upper left corner of a bounding box, and *endX* and endY correspond to the x- and y-coordinates of the lower right corner of a bounding box. In the first frame of the video, each bounding box is assigned a unique ID by the function register. This function stores the IDs as keys and the bounding box coordinates as box coordinates as values in the dictionary objects. In each subsequent frame *n*, the bounding boxes are mapped to the bounding boxes of the previous frame *n* − 1 using the Euclidean distance *d*. Here, the Euclidean distances between the centroids of each pair of bounding boxes in frame *n* and frame *n* − 1 are computed and stored in a matrix $M_{dist}$, where the columns correspond to the bounding boxes of frame *n* − 1 and the rows correspond to the bounding boxes of frame *n*:$$M_{dist}=\left(\begin{array}{ccc} d\;({centroid\;cell\;1}_{n-1},{centroid\;cell\;1}_{n}) & d\;({centroid\;cell\;2}_{n-1},{centroid\;cell\;1}_{n}) & \ldots \\ d\;({centroid\;cell\;1}_{n-1},{centroid\;cell\;2}_{n}) & d\;({centroid\;cell\;2}_{n-1},centroid\;{cell\;2}_{n}) & \ldots \\ \ldots & \ldots & \ldots \end{array}\right)$$

Additionally to the tracking proposed by Rosebrock [22], the IoU overlap of each box in frame *n* − 1 and frame *n* was calculated and also stored in a matrix $M_{IoU}$ similar to $M_{dist}$. A matching matrix was computed using the following:$$M_{matching}=M_{dist}+0.1-0.1\;\cdot \;M_{IoU}$$

Now, the algorithm searches each row of the matching matrix for the smallest value. These values are ordered in ascending order, and the position of each value in the matrix is also stored. Starting with the first and smallest value, the corresponding bounding boxes in frame *n* and frame *n* − 1 are derived based on the position in the matrix. It can be assumed that these two bounding boxes contain the same object since they are the closest to each other and overlap the most. Therefore, the ID of the bounding box in frame *n* − 1 is assigned to the bounding box in frame *n*. This step is repeated for all the smallest values in all rows. To avoid double assignment of the bounding boxes of both frames, the indices of the bounding boxes used are stored. Before each new assignment of an ID, it is checked whether the bounding box in frame *n* − 1 has already been matched to a bounding box of frame *n* or vice versa. Each time a bounding box is matched, the dictionary objects are updated by assigning the value tuple containing the coordinates of the matched bounding box of frame *n* to the key ID that was found to be the matching ID. Finally, it is checked whether there are bounding boxes in frame *n* or *n* − 1 that have not been matched to a box of another frame. If a bounding box of frame *n* has no matching partner, the function register is executed. If there is no matching partner for a bounding box in frame *n* − 1, the deregister function is executed. This function deletes the ID and coordinates of this bounding box from the dictionary objects. This procedure is repeated for each frame of the video. By using the matching matrix instead of the Euclidean distance matrix for the matching, the matching of two boxes that do not overlap or only partly overlap is penalized, and the matching of boxes that have a small center-to-center distance and a huge box overlap is encouraged.

### 4.5. Tracking Accuracy

The tracking accuracy was measured by applying the centroid tracker to the groundtruth-bounding boxes of the two test videos and comparing the IDs assigned to the boxes by the tracker to the groundtruth IDs. As the centroid tracker is not able to track across cell divisions, the specific ID changes at cell divisions were ignored for the scoring, and only the consistency of the tracks in between cell divisions was scored. The IDs of each object in a frame *n* > 0 were compared to the ID of the very same object in the previous frame *n* − 1 in the tracks created by the centroid tracker (prediction) and the tracks created by me (groundtruth). If the ID of an object did not change in the groundtruth and the prediction between frame *n* and frame *n* − 1, the variable *right_ID,* which was initially 0, was increased by 1. If the ID of an object changes in the prediction but not in the groundtruth, the variable *wrong_ID*, which was initially 0, was increased by 1. This was performed for all bounding boxes and frames in the video. Afterwards, the tracking accuracy *t_acc_* was calculated by using the following formula:$$t_{acc}=\frac{right\_ID}{right\_ID+wrong\_ID}$$

### 4.6. Cell Culture and Irradiation

CHO cells were cultivated in the growth medium RPMI 1640 (R8758-500ML, Sigma-Aldrich, St. Louis, MO, USA) supplemented with 10% FCS (fetal calf serum, F0804-500ML, Sigma-Aldrich, USA), 1% Penicillin-Streptomycin (P4333-100ML, Sigma-Aldrich, USA), and 1% Sodium Pyruvate (S8636-100ML, Sigma-Aldrich, USA). Cells were grown in an incubator at a temperature of 37 °C, 5% CO_2_, and 100% humidity and were passaged twice a week. Cells were seeded for 24 h before irradiation on µ-dishes with glass bottoms (μ-Dish 35 mm, Ibidi, Martinsried, Germany). For irradiation, an X-ray cabinet (CellRad, Precision Xray Inc., Madison, CT, USA) was used. One dish was irradiated with 3 Gy of X-rays (130 kV) and a dose rate of 0.067 Gy/s and the other dish was sham irradiated.

### 4.7. Life-Cell Microscopy

Cells were imaged by live-cell microscopy for up to 4 days. Therefore, the microscope was equipped with a stage-top incubator (Tokai-hit STX, Tokai-hit, Fujinomiya, Japan). The incubator allows cells to be maintained under cell culture conditions (37 °C temperature, 5% CO_2_ concentration, and 100% humidity). The water reservoir in the stage-top incubator was replenished every day during the recording period, and the growth medium for the cells was replenished every second day. The water reservoir, high humidity, and medium replenishment prevented the cell sample from drying out and ensured optimal physical conditions and nutrient supply for the cells during observation. A 10× objective (Plan-Apochromat 10×/0.45 Ph1, Zeiss, Oberkochen, Germany) was used for imaging. Cells were imaged in two modes: in standard phase-contrast using the phase stop Ph 1 (Zeiss, Germany) as suggested by the manufacturer and hereafter referred to as mode 1. In addition, videos were recorded in a bright-field phase-contrast mode using the phase stop Ph 2 (Zeiss, Germany), hereafter referred to as mode 2. Both imaging methods were included in the dataset to increase the robustness of the detection algorithm and to encourage the algorithm to learn the structure and pattern of different cell morphologies rather than intensity patterns. The test videos were recorded in mode 1.

## 5. Conclusions

We show that CeCILE 2.0 can be used to accurately detect and track cells in phase-contrast videos in a human-supervised mode. The results provided by CeCILE 2.0 provide information on cell growth, cell cycle duration, first cell divisions, proliferation, cell cycle abnormalities, relationships between cells in a sample, and cell survival. Thus, CeCILE 2.0 combines all important endpoints of radiobiological research and provides more detailed information that cannot be derived from state-of-the-art assays, such as cell cycle abnormalities. In addition, CeCILE 2.0 could achieve significant results by analyzing only small numbers of cells. Gold standard assays such as the colony-forming assay require several thousand cells to obtain significant results [14]. No cell staining or specific treatment is required, as is the case with CFA or the Caspase3/7-Sytox assay [16], which evaluates cell death. Possible effects of treatments can be avoided by using CeCILE 2.0. Since the cells are monitored every 5 min, detailed information about the cells can be derived for each time point. Thus, a detailed history of each cell is obtained. State-of-the-art assays are typically evaluated at specific time points. The CFA is typically evaluated 5 days after irradiation and the Caspase3/7-Sytox assay at 24 h in the case of CHO-K1 cells. If information at other time points is of interest, these assays need to be repeated. Currently, CeCILE 2.0 is used in a human-monitored mode, which we aim to overcome in the next development step of CeCILE using different tracking approaches.

## Figures and Tables

**Figure 1 cells-12-02782-f001:**
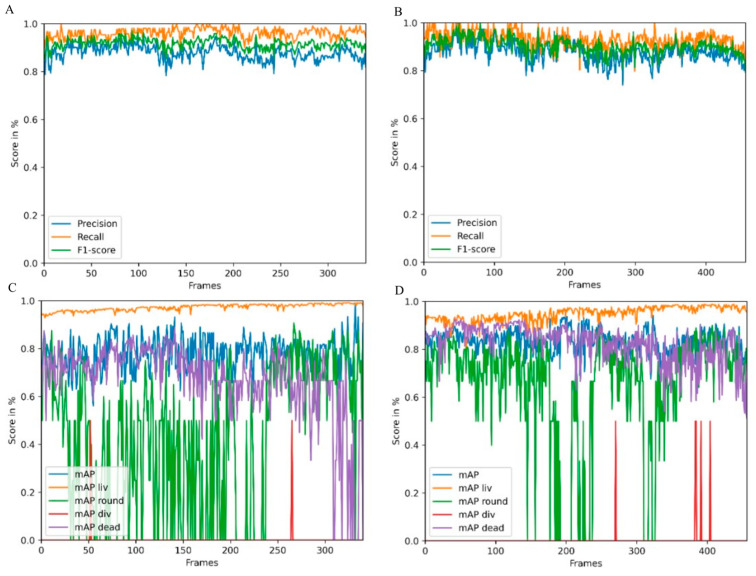
Accuracy of object detection in a video of irradiated (**A**) and non-irradiated (**B**) cells over 340 frames. Shown are precision in blue, recall in orange, and F1-score in green. (**C**,**D**) show the overall mean average precision (mAP) and the mAP for each class (liv: living cells, round: round cells, div: dividing cells, dead: dead cells) for the irradiated (**C**) and the non-irradiated sample.

**Figure 2 cells-12-02782-f002:**
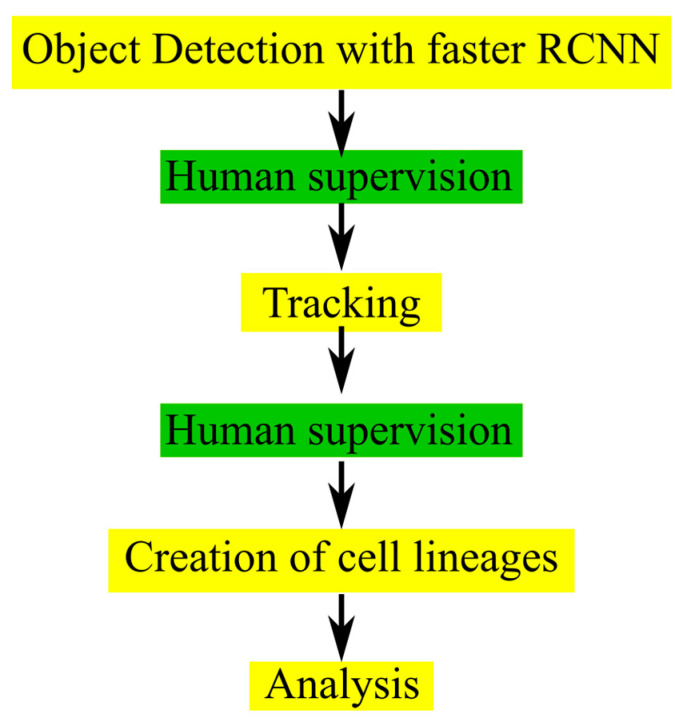
Chart of the workflow used in CeCILE 2.0. The steps with human interaction are shown in green, and the fully automated steps are shown in yellow.

**Figure 3 cells-12-02782-f003:**
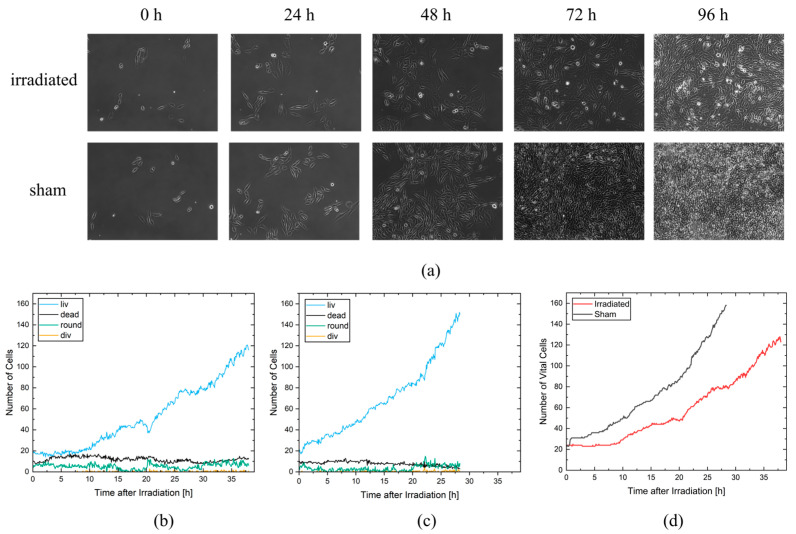
(**a**) Phase-contrast images of the irradiated and non-irradiated sham cell population. (**b**) Number of cells per class for the non-irradiated sham cell population. (**c**) Number of cells per class for the irradiated cell population. (**d**) Comparison of all vital cells for irradiated (red) and non-irradiated sham (black) cell populations.

**Figure 4 cells-12-02782-f004:**
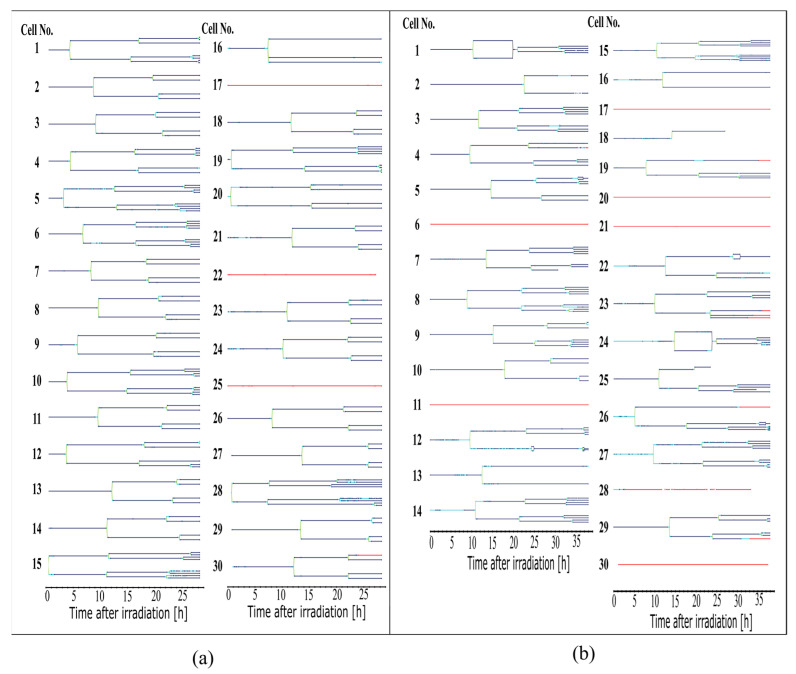
Cell lineages of all cells evaluated in the sham-irradiated control (**a**) and the 3 Gy X-ray-irradiated sample (**b**). Lines in the lineages appear dark blue for living cells, light blue for round cells, green for dividing cells, and red for dead cells. Black connections between two cells indicate a fusion of the membranes of two connected cells.

**Figure 5 cells-12-02782-f005:**
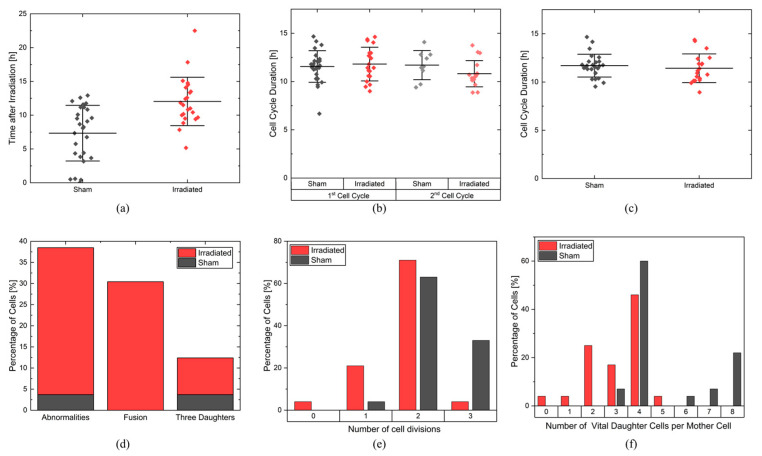
(**a**) Time until first cell division. Each cell is depicted with a data point for a sham (black) and an irradiated sample (red). The mean and the corresponding standard deviation are shown. (**b**) Cell cycle duration for the 1st and 2nd complete cell cycles for both sham (1st cell cycle: black, 2nd cell cycle: gray) and irradiated (1st cell cycle: red, 2nd cell cycle: light red). The mean and standard deviation are shown. (**c**) The mean cell cycle duration for all fully imaged cell cycles in the sham (black) and irradiated (red) samples are shown. The mean cell cycle duration for each cell is depicted in each data point. The mean and standard deviation for each population are shown. (**d**) The percentage of cells showing one or more cell cycle abnormalities, showing fusion, and showing division in three daughter cells during recording are depicted. Red: irradiated population. Black: non-irradiated sham population. (**e**) The percentage of initial cells showing a certain number of divisions in 28.3 h after irradiation. Red: irradiated population. Black: non-irradiated sham population. (**f**) Percentage of initial cells having a certain amount of vital daughter cells 28.3 h after irradiation. Red: irradiated population. Black: non-irradiated sham population.

**Figure 6 cells-12-02782-f006:**
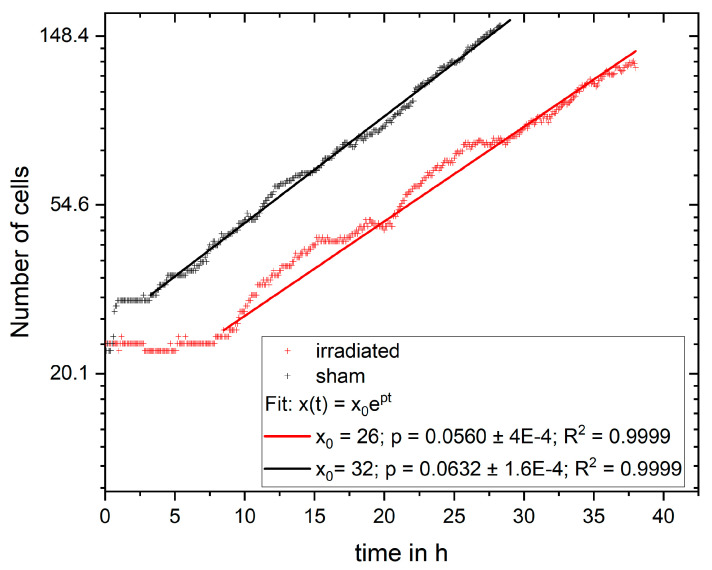
Number of cells on logarithmic axis versus time with the fit of exponential growth.

**Table 1 cells-12-02782-t001:** Summary of the mean, maximum, and minimum values of precision, recall, and F1-score for the irradiated and non-irradiated video.

	Irradiated	Non-Irradiated
Mean precision	0.88	0.88
Max precision	0.98	0.92
Min precision	0.78	0.80
Mean recall	0.93	0.96
Max recall	1.0	1.0
Min recall	0.82	0.95
Mean F1-Score	0.90	0.92

**Table 2 cells-12-02782-t002:** Comparison of expected and measured values for the number of cell divisions (Div) and the number of daughter cells (DC) for irradiated and non-irradiated samples.

	Irradiated	Non-Irradiated
p	$0.0560\pm 4{\times 10}^{-4}$	$0.0632\pm 1.6\times 10^{-4}$
DC_calc_	3.1 ± 0.9	5.0 ± 0.7
DC_meas_	3.1 ± 0.25	5.1 ± 0.4
Div_calc_	1.6 ± 0.5	2.5 ± 0.4
Div_meas_	1.75 ± 0.13	2.29 ± 0.11

**Table 3 cells-12-02782-t003:** Parameters used for object detection training.

Parameter	Value
Number of epochs	100,000
Learning rate	Cosine decay learning rate, with
	warm-up (3000 epochs): 0.001
Aspect ratio	0.5, 1.0, and 2.0
Scales of the anchor boxes	0.25, 0.5, 1.0 (256 × 256 pixels), and 2.0
Data augmentation	Horizontal flip, adjust brightness, contrast, square crop by scale
Non-maximum suppression	IoU: 0.3 and 0.35
Localization loss weight	3 and 2
Classification loss weight	1
Objectness loss weight	1.5

## Data Availability

The data presented in this study are available on request from the corresponding author. The data are not publicly available due to restrictions set by the university.

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
