# Peer review of "Single-Cell Radiation Response Scoring with the Deep Learning Algorithm CeCILE 2.0"

_cells, 2023, doi:10.3390/cells12242782_

Round 1
Reviewer 1 Report
Comments and Suggestions for Authors
The study focused on the development of AI-based approach for analyzing cell growth, cell cycle duration, first cell divisions, proliferation etc. The model was trained on recorded videos of cell growth at different time points. A literature review is well presented in the introduction but needs little improvement from a hypothesis-building point of view. The results and discussion are written well. Methods section needs more clarity. I have the following concerns-
1) Paragraph “Additionally, radiation can induce oxidative stress by generating reactive oxygen species s….” needs more references. I would recommend adding citations for each biological phenomena individually.
2) Do we have any centralized resource for data on “live-cell phase-contrast videos”. It would be informative if the authors can introduce some of these in the introduction.
3) Was “maximum number of cells” the only limitation in the previous version of CeCILE? I think this is the main backbone for hypothesizing the current study. I would recommend adding more text and discussion about the limitations and current hypothesis in the introduction section.
4) Para “The predictions were compared to a groundtruth with manually corrected prediction videos..”, Can you please elaborate this? Were predictions compared with another prediction? What is ground truth here?
5) Please expand mAP liv, mAP div etc in the Figure 1 legend/caption.
6) What is the source of videos of HeLa, CHO, LN229 cells? Please do mention in the method section.
7) The formulation of M(matching) is not explained. What are the optimization steps of this equation?
8) A figure illustrating a flow chart of methodology would be very helpful for the readers.
9) What is the main findings in Figure 3? Please explain.
10) Figure 4e and f, any idea why % cells (both radiated and non-radiated) decrease after a certain number of divisions?
11) What are the limitations of the CeCILE 2? I recommend adding in the text.
12) Are there any existing methods other than CeCILE? Authors should consider those to compare.
Author Response
Dear reviewer,
we want to thank you for providing a detailed and constructive review. We tried to adress all the comments made. Please find the detailed answers in the attached document.

Reviewer 2 Report
Comments and Suggestions for Authors
Abstract
- Consider clarifying the unique contributions of CeCILE 2.0 compared to previous versions or similar algorithms in the field. It's essential to highlight what sets this version apart to capture the reader's interest.
Introduction
- The transition between discussing general radiation effects on cells and the specific focus of this study could be smoother. A suggestion would be to add a sentence that directly links the general impact of radiation to the need for advanced tools like CeCILE 2.0 for detailed analysis.
4.2 Object Detection
- It's stated that the Faster RCNN is "easy to use and to adapt to custom datasets" and is "very accurate in detecting many small objects on a crowded image." While this provides a general understanding, it would be beneficial to include specific examples or data demonstrating this adaptability and accuracy, especially in the context of cell imaging.
- The use of transfer learning with a pretrained ResNet-101 model is mentioned, but the rationale for choosing this specific model and its effectiveness in this particular application could be elaborated upon. Providing insights into why ResNet-101 was selected and how it contributes to the overall performance of CeCILE 2.0 would be valuable.
- It would be beneficial to explicitly state whether the layers of the pretrained ResNet-101 model were frozen during the training of CeCILE 2.0. Freezing layers is a notable technique in transfer learning, particularly when employing a complex model like ResNet-101.
- The manuscript mentions a 75%/25% split for training and validation datasets, which is standard, but additional details about the dataset size and characteristics would enhance the reader's understanding. Furthermore, elaborating on the criteria for evaluating the object detector’s predictions during training could provide more insight into the validation process.
- While referencing Rosebrock and the TensorFlow 2 object detection API for the data preparation and training pipeline is helpful, a brief summary of key steps or modifications made specific to CeCILE 2.0 would be informative. This would illustrate the custom approach taken for this specific application.
- The section lists various parameters that were finetuned during training, which is crucial for understanding the model's configuration. However, some readers may benefit from a brief explanation of why each parameter was chosen for fine-tuning and how adjustments to these parameters impact the model's performance. Additionally, directing the reader to where they can find the final parameters in Table 3 is helpful, but a brief discussion of the most impactful parameters in the text could be advantageous.
Results
- To fully appreciate the advancements and improvements made in CeCILE 2.0, it would be beneficial for readers to see these metrics directly compared with those of CeCILE 1.0.
Overall, the manuscript presents interesting and valuable research. Enhancing these sections would strengthen the paper's impact and clarity, providing a more comprehensive understanding of the algorithm's capabilities and limitations.
Author Response
Dear reviewer,
we thank you for your very valuable comments. We tried to adress all of the points in detail. Please find the answers in the attached document.

Round 2
Reviewer 1 Report
Comments and Suggestions for Authors
Thanks for the revisions. All the comments were addressed. Just a quick suggestion-
1) You should mention line numbers in the manuscript draft and in all the revisions.
2) Please mention the line number in the cover letter for any changes/edits you make in the manuscript.
This is helpful for the reviewers to track the changes and revisions.
Comments on the Quality of English LanguageMinor editing is required.